

# Baltic Sea shipwrecks as a source of hazardous pollution

Agata Szpiech[1], Aleksandra Bojke[1], Małgorzata Littwin[1], Aleksandra Dzendrowska[1], Ewelina Duljas[1], Agnieszka Flasińska[1], Katarzyna Szczepańska[1], Tomasz Dziarkowski[1], Grażyna Dembska[1], Grażyna Pazikowska-Sapota[1], Katarzyna Galer-Tatarowicz[1] and Benedykt Hac[2]

[1] Department of Environment Protection, Maritime Institute, Gdynia Maritime University, Gdynia, Poland
[2] Mare Foundation, Warszawa, Poland

## ABSTRACT

**Background**. Shipwrecks on the Baltic Sea seabed pose a serious threat to the marine environment. Fuel, ammunition and chemicals in their holds can enter the ecosystem at any time, causing an ecological disaster. It is known that oil spills from ship accidents can affect life and health of different species of animals, both immediately after catastrophe and for many years thereafter. This article discusses the negative impact of shipwrecks on the ecological status of the Baltic Sea and presents the contamination status of bottom sediment core samples taken in the vicinity of shipwrecks located in the South Baltic, *i.e.*, S/s Stuttgart, t/s Franken, S/T Burgmeister Petersen and m/s Sleipner. It is based on the results of research carried out by the Maritime Institute between 2011 and 2016.

**Methods**. Core samples were taken by a VKG-2 vibrating probe and analysed towards content of polycyclic aromatic hydrocarbons (PAHs), polychlorinated biphenyls (PCBs), total petroleum hydrocarbons (TPHs) and total organic carbon (TOC). Seven PAHs and Seven PCBs were determined using solid phase extraction (SPE) technique followed by gas chromatography-mass spectrometry (GC-MS) detection. In order to determine the concentration of TPH, the SPE technique was applied followed by gas chromatography with flame ionisation detector (GC-FID) analysis. TOC content was established by TOC analyser using infrared detection.

**Results**. Samples taken in the vicinity of the S/s Stuttgart shipwreck have shown the highest concentrations of all analysed parameters compared to the examined wrecks and significantly differed from the results typical for these areas. This studies have shown that the S/s Stuttgart shipwreck poses a serious threat to the marine environment of Baltic Sea and it is necessary to continue studies in this area and to perform a wider range of analysis.

Corresponding author
Aleksandra Bojke,
abojke@im.umg.edu.pl

## INTRODUCTION

The Baltic Sea is an inland water body with an area of 415,266 km$^2$ characterised by low salinity compared to oceans. This is caused, *e.g.*, by the large amount of river run-off, limited water exchange with the Atlantic Ocean and the predominance of precipitation over evaporation. These specific conditions mean that the Baltic Sea has a tendency

to accumulate pollutants, as a result of which a progressive increase in environmental deterioration processes has already been observed since the 1950s (*Andrulewicz et al., 1994*; *Andrulewicz et al., 1998*; *Dobrzycka-Krahel & Bogalecka, 2022*). The factors currently most threatening the Baltic Sea environment are excessive eutrophication, nutrient influx, loss of biodiversity and the presence of waste (*HELCOM, 1974*, *HELCOM. 2018*, *Jedruch et al., 2023*). These result in an increased presence of phytoplankton and reduced oxygen concentrations in the water, which in turn negatively affect the flora and fauna (*Murray et al., 2019*). Increased biogenic content in the Baltic Sea is largely related to anthropogenic activities conducted by industrial plants, sewage treatment plants, agriculture or animal husbandry (*Andrulewicz et al., 1998*; *Bogalecka & Kołowrocki, 2016*). In addition to biogenic substances, rivers and canals also carry other pollutants into the sea in the form of various toxic substances, *e.g.*, heavy metals, organic compounds, detergents and plant protection products (*Łomniewski, Mańkowski & Zaleski, 1975*; *Szafrańska, Gil & Nowak, 2021*). Moreover, there is an increased content of oil-based pollutants in port and shipyard areas and their adjacent zones, as well as on shipping routes (*Andrulewicz et al., 1994*; *Ekere et al., 2019*). Waste discharges, fuel spills, shipping, raw material extraction and marine accidents are the main causes of marine pollution.

However, there are other sources of pollution that are not often mentioned but pose a serious threat to the entire aquatic ecosystem (*Sprovieri et al., 2013*, *Soares et al., 2020*). These are sunken warships and merchant ships abandoned on the seabed, which can cause ecological damage as they very often have oil, chemicals or weapons on board (*Monfils, Gilbert & Nawadra, 2006*). Shipwrecks can also locally alter the topography of the seabed, the direction and speed of sea current, as well as affecting nutrient and oxygen concentrations (*Balazy, Copeland & Sokołowski, 2019*). Oil, depending on its origin, can take the form of light oil, heavy oil, bitumen or intermediate forms, which differ in the ratio of light and heavy hydrocarbons. In an oil spill situation, the chemical composition of oil will determine the impact it will have on the marine environment (*Zhang et al., 2019*). Studies conducted off the coast of Alaska following the 1989 MT Exxon Valdez catastrophe proved that oil spills not only cause a sharp increase in animal mortality, but also, through long-term exposure to toxic substances, adversely affect the health, growth and reproduction of organisms in subsequent years (*Peterson et al., 2003*). Similar conclusions were drawn from studies performed off the coast of Spain following the MT Prestige sinking. Studies conducted several months after the accident on gulls living in the affected areas confirmed the presence of damage to the health of these animals (*Alonso-Alvarez et al., 2007*).

Polycyclic aromatic hydrocarbons (PAHs) are the ingredients of the oil and one of the ways they enter the environment is through ship accidents and oil spills (*Ambade et al., 2021*). Due to high hydrophobicity and low solubility of PAHs their main place of accumulation are sediments (*Jesus et al., 2022*). Furthermore phytoplankton and marine organisms have the ability to bioaccumulate PAHs (*Duran & Cravo-Laureau, 2016*). It is known that some PAHs may have carcinogenic, mutagenic or teratogenic effect (*Mallah et al., 2022*) and can cause, for example diabetes, infertility or cardiovascular disease (*Gao et al., 2018*). According to the U.S. Environmental Protection Agency (EPA) there

are seven identified human carcinogenic PAHs: benzo(a)anthracene, benzo(a)pyrene, benzo(b)fluoranthene, benzo(k)fluoranthene, chrysene, dibenzo(a)anthracene, and indeno(1,2,3-c,d)pyrene (*Banger et al., 2010*).

The protection and monitoring of the Baltic Sea environment is primarily the responsibility of the international organisation the Baltic Marine Environment Protection Commission (Helsinki Commission, HELCOM). HELCOM is the implementing body of the Helsinki Convention originally drawn up in 1974, which was signed by all Baltic Sea coastal states. HELCOM experts continuously analyse the state of the environment and the presence of pollutants in the Baltic Sea and produce reports and recommendations for the Member States, which are obliged to take measures for the protection of the Baltic Sea basin. During the HELCOM conference in 2021, the topic of the threat to the marine environment posed by shipwrecks and weapons lying on the Baltic Sea seabed was discussed (*HELCOM, 2021*). In 1948, a specialised agency of the United Nations—the International Maritime Organisation (IMO)—was founded. The IMO is responsible for, *e.g.*, maritime safety and the prevention of pollution of the marine environment by ships. The IMO's activities are reflected in the numerous international conventions that have been enacted. One of the most important documents addressing the problem of shipwrecks lying on the seabed in marine areas is the Nairobi International Convention on the Removal of Wrecks from 2007. This document entered into force in 2015 and has been ratified by dozens of countries from around the world. Among other things, this Convention describes procedures for reporting, locating, marking and removing wrecks as well as determining the level of danger they pose (*International Maritime Organization, 2007*).

In Poland, to date, no legal regulations have been introduced to address the problem of hazardous shipwrecks lying on the Baltic Sea seabed, nor have any systems been developed to counteract this threat. However, since 2011, the Chief Inspectorate for Environmental Protection (CIEP) has been regularly assessing the state of marine waters, and in 2020, for the first time, it analysed the waters in terms of risks from the presence of dumped chemical weapons and shipwrecks (*Chodkiewicz et al., 2021*). The topic of sunken shipwrecks in the Baltic Sea and the associated risks has been popularised by the NGO The MARE Foundation, founded in 2016. Its aim is to protect marine ecosystems, and its activities include the introduction of wreck management programme in Poland (*Fundacja MARE, 2023*).

The highly developed maritime transport and warfare of the past century has left tens of thousands of sunken ships on the bottom of the seas and oceans (*Ndungu et al., 2017*). It is estimated that nearly 9,000 oil tankers and other crafts, capable of holding more than 20 million tonnes of oil, have sunk in waters around the world (*Michel et al., 2005*). In the Gulf of Gdańsk alone, 45 sunken metal and wooden shipwrecks have been found at various depths and of various sizes, from small fishing boats to large commercial units (*Jeffery, 1990*; *Lastumäki et al., 2020*; *Vieira et al., 2023*). While wooden shipwrecks provide an interesting site for recreational diving, steel ones with oil cargo are potentially very dangerous objects, posing a threat to the environment. It is estimated that the size of all wrecks in this area is 0.01 km$^2$ (*Sokołowski et al., 2021*).
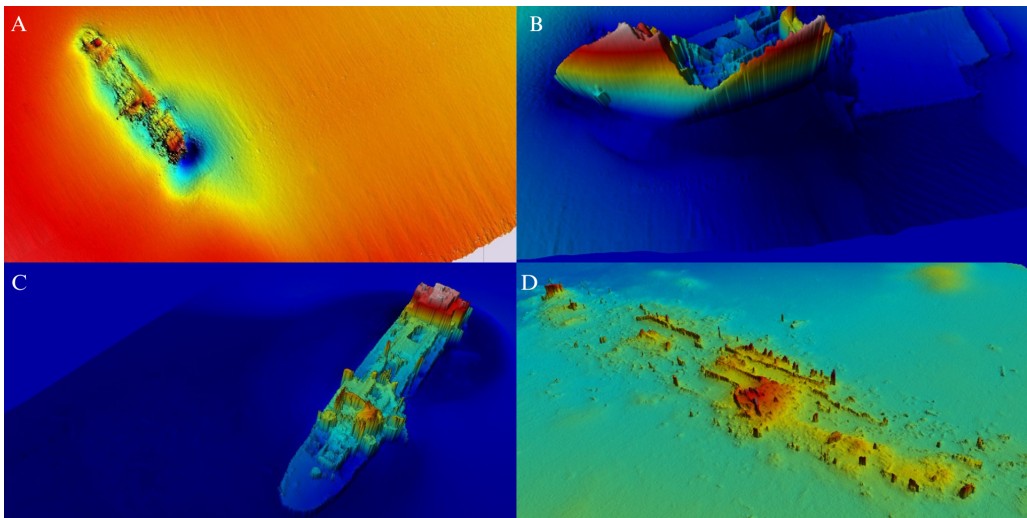

**Figure 1 Bathymetric charts of shipwreck locations at the Baltic Sea seabed.** (A) S/T Burgmeister Peteresen; (B) m/s Sleipner; (C) t/s Franken (*Hac, 2016*); (D) S/s Stuttgart (*Hac, 2021*). Image credit: Maritime Institute.

North of Jastarnia (Poland) lies the wreck of the S/T Burgmeister Petersen, a tanker built in 1889 and used to transport liquids. In 1915 this tanker hit a mine, which caused her to explode and break in half (Fig. 1A). It can be assumed that at the time of the sinking, the craft may have had more than a thousand or so tonnes of light fuel in her tanks, which was used to supply submarines, and more than a hundred tonnes of heavy fuel, which was used to fuel the ship's main boiler (*Underwater.pl, 2023*; *Dembska et al., 2011*). Another shipwreck that rests off the Polish coast is the m/s Sleipner. She was built in 1965 and in 1975 collided with another ship and ran aground with a torn side. A month later, her hull broke in half during a storm (Fig. 1B) (*Hac et al., 2016b*; *Hac, 2016*). Another shipwreck located in the Gulf of Gdańsk that poses a potential threat to the environment is t/s Franken. The ship was built in 1943 and had 13 cargo tanks in which she was able to transport various petroleum products such as heavy fuel, light fuel and engine oils. Franken was sunk in 1945 as a result of a bomb attack. At the time of her sinking, she may have had around 3,000 tonnes of fuel in the tanks. It is estimated that today between 300 and 600 tonnes of fuel may still be inside this shipwreck (Fig. 1C) (*Hac et al., 2016a*; *Hac, 2018*). The environmentally hazardous shipwreck of S/s Stuttgart is also located in the Gulf of Gdańsk. She was built in 1923 as a passenger craft and later converted into a hospital ship. In 1943, the S/s Stuttgart, while moored in the port of Gdynia, was bombed, resulting in a fire on board. Due to the danger posed by the ship on fire, she was towed away from the shore and dumped. The shipwreck of the S/s Stuttgart lying north of the entrance to the port of Gdynia posed a navigational hazard and in 1953 a decision was taken to dismantle her by pyrotechnic destruction, which resulted in the contamination of the seabed area around the ship (Fig. 1D) (*Rogowska, Wolska & Namieśnik, 2010*).

Since 1999, the Maritime Institute in Gdańsk (since 2019 the Maritime Institute of the Gdynia Maritime University) has been conducting research on shipwrecks resting on the Baltic Sea seabed to assess their impact on the marine environment and to determine the threat they may pose. This publication presents the historical results of core bottom sediment chemical analyses taken in the vicinity of four shipwrecks, S/T Burgmeister Petersen, m/s Sleipner, t/s Franken and S/s Stuttgart, which are lying on the bottom of the Baltic Sea.

## MATERIALS & METHODS

### Location of shipwrecks

Surveys were carried out around four shipwrecks located within Polish territorial waters of the Baltic Sea: S/T Burgmeister Petersen, m/s Sleipner, t/s Franken and S/s Stuttgart. Their exact locations are shown in Fig. 2. Bottom sediment core samples were taken using a VKG-3 vibrating probe with an inner tube diameter of approximately 10 cm, located on board the Institute's research vessel. In determining the sampling locations, the staff of the Maritime Institute considered the catastrophe history of each shipwreck, the current location or position of individual parts of each shipwreck, the shape of the bottom in their vicinity and the analysis of currents and wave action. Due to the threat posed by the presence of the contents of their tanks in the marine environment, great importance was given to the design of the ships with particular reference to the location of the fuel as well as bilge tanks.

### Location of sampling points

Four sampling sites were mapped around the S/T Burgmeister Petersen shipwreck (Fig. 3), five around the m/s Sleipner shipwreck (Fig. 4), five around the t/s Franken shipwreck (Fig. 5) and 20 stations around the S/s Stuttgart shipwreck (Fig. 6). In the case of S/s Stuttgart, the sampling points were both in places where moderate and lesser spill was expected as well as outside the area of expected spill (Fig. 6). The sampling in the vicinity of S/T Burgmeister Petersen took place in 2011 and the other samplings in 2016. The bottom sediment core samples taken in the vicinity of the S/T Burgmeister Petersen ranged from 2.00 m to 2.35 m in length, the samples taken in the vicinity of m/s Sleipner ranged from 2.70 m to 3.00 m, and the samples taken in the vicinity of the t/s Franken ranged from 1.70 m to 3.00 m. For the area around the S/s Stuttgart, a core of 1.00 to 2.70 m was sampled at each of the designated stations.

### Laboratory analysis

After sampling, sediments were stored in fridge in $4 \pm 3$ °C until freeze-drying. All analysis were performed in the accredited Environmental Protection Laboratory (PCA Certificate, the 646). Quality assurance was achieved by performing analysis of blank samples, duplicate samples, samples spiked with analytical standard and analysis of Certified Reference Materials (*e.g.*, TPH-Sandy Loam 3, Nutrients in Sandy Loam, PAH-Sediment 1, PCB congeners—Loamy Sand produced by Sigma-Aldrich, River Sediment—PAHs produced by LGC). Quality guarantee was checked in Proficiency Testing organised by LGC (CONTEST).

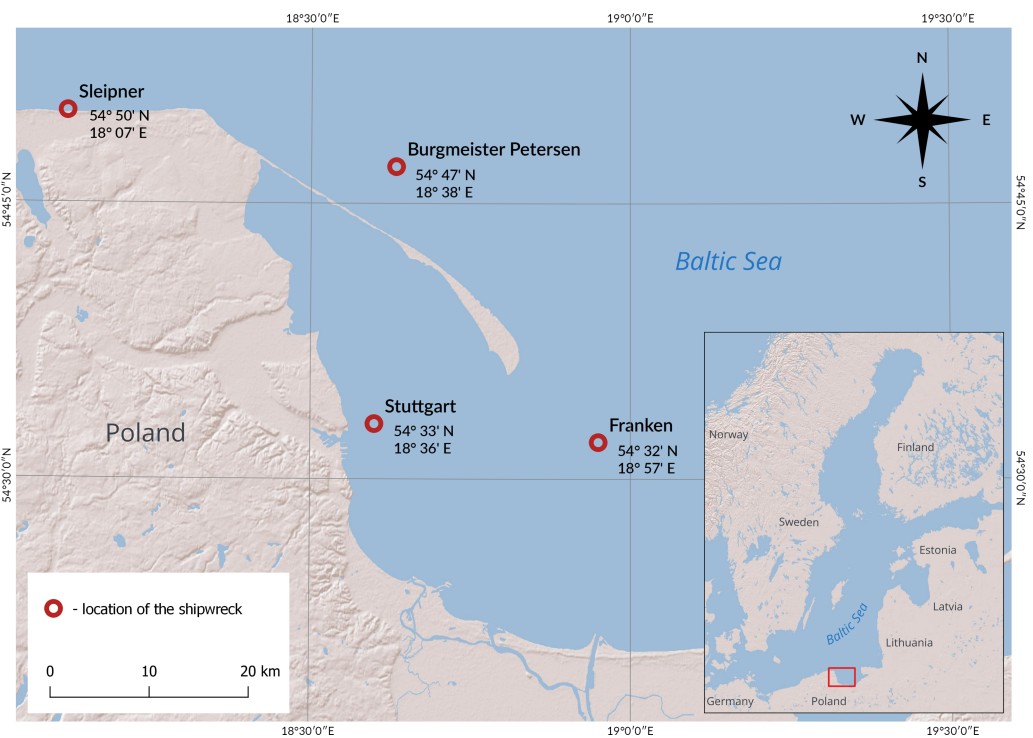

**Figure 2** **Chart showing locations of surveyed shipwrecks.** Base Map credit: Copyright:©2014 Esri (Redlands, CA, USA).

## Determination of PAHs and PCBs

The sediment samples collected for the study were analysed for seven PAHs: benzo[a]anthracene, benzo[b]fluoranthene, benzo[k]fluoranthene, benzo[a]pyrene, indeno[1,2,3-cd]pyrene, dibenzo[a,h]anthracene and benzo[ghi]perylene and seven polychlorinated biphenyls (PCBs): (PCB28, PCB52, PCB101, PCB118, PC138, PCB153 and PCB180) selected in accordance with the legal acts in force in Poland (*Journal of Laws, 2015*). Extraction of the above compounds was performed by shaking the sample with dichloromethane. The solid phase extraction (SPE) technique was used to purify the extract using extraction columns packed with silica gel (JT Baker; silica gel; 1,000 mg) and hydrochloric acid-activated copper. All columns were conditioned with hexane prior to extraction. The purified extract was concentrated and analysed by gas chromatography-mass spectrometry detection using a GC-MS instrument (GC: Hewlett Packard 6890; Hewlett Package, Palo Alto, CA, USA; MS: Hewlett Packard 5973). The limit of quantification (LOQ) of this method for individual as well as the sum of PAHs was 0.001 mg/kg d.w., and for individual congeners and the sum of PCBs 0.0001 mg/kg d.w.

## Determination of TPH

To analyse the content of total petroleum hydrocarbons (TPH) (C10-C40), samples were subjected to ultrasound-assisted extraction with hexane, followed by further extraction assisted by mechanical stirring. The solid-phase extraction (SPE) technique was used to

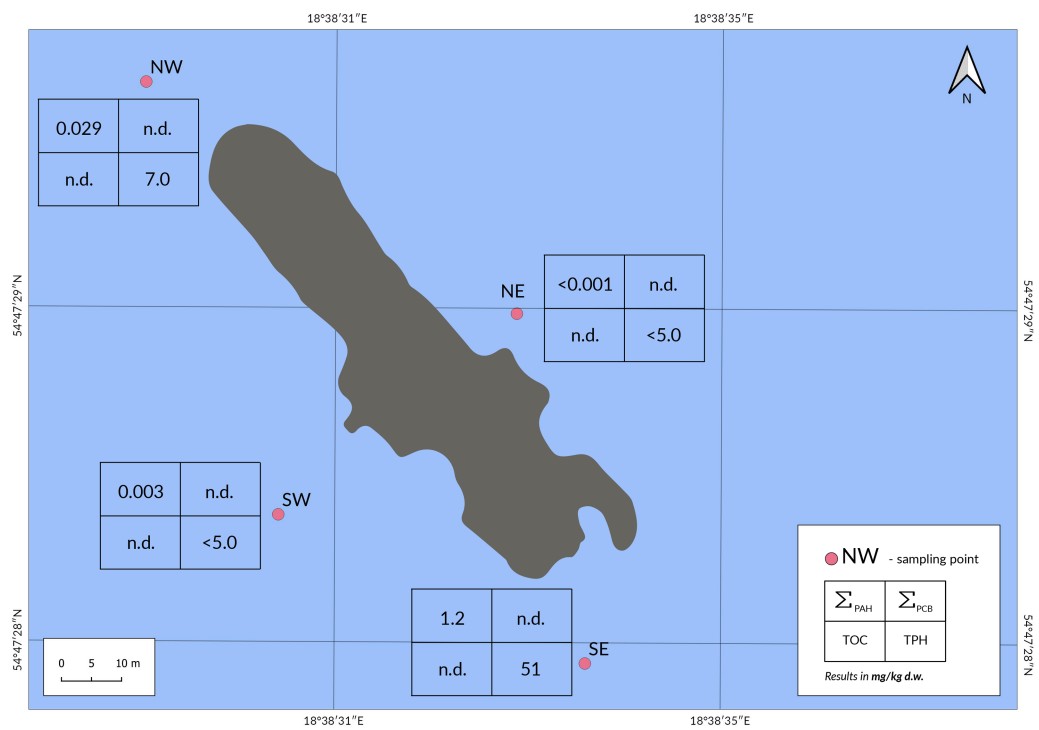

**Figure 3** **Location of the sampling points in the Burgmeister Petersen shipwreck area, together with the results obtained for the parameters tested.** Abbreviation: n.d., no data. Image credit: Karolina Duljas.

purify the extract using extraction columns containing aluminium oxide and anhydrous sodium sulphate. The purified extract was analysed by gas chromatography with a flame ionisation detector (GC-FID: Agilent Technologies 6890N; Santa Clara, CA, USA). The LOQ using this method was 5.0 mg/kg d.w.

## Determination of TOC

To determine the content of total organic carbon (TOC), a TOC analyser using infrared detection was used (TOC-L instrument with SSM-5000A adapter for the analysis of solid samples made by Shimadzu, Kyoto, Japan). The LOQ using this method was 1,000 mg/kg d.w.

## QGIS software

In developing the charts with sampling points around the shipwrecks, QGIS version 3.10 software was used (https://qgis.org/en/site/). The 1992 coordinate system (EPSG: 2180) was applied.

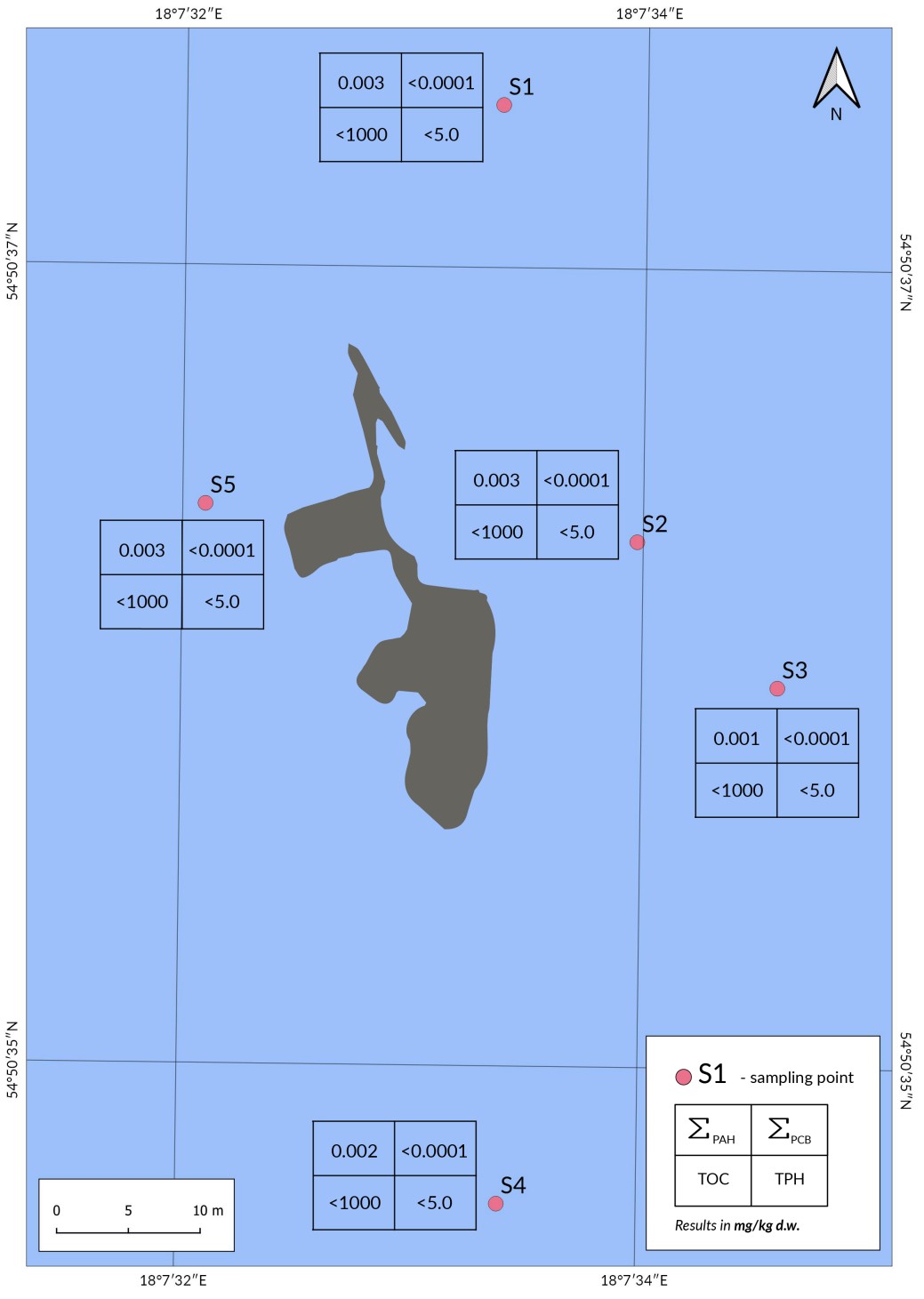

**Figure 4** **Location of the sampling points in the Sleipner shipwreck area, together with the results obtained for the parameters tested.** Image credit: Karolina Duljas.

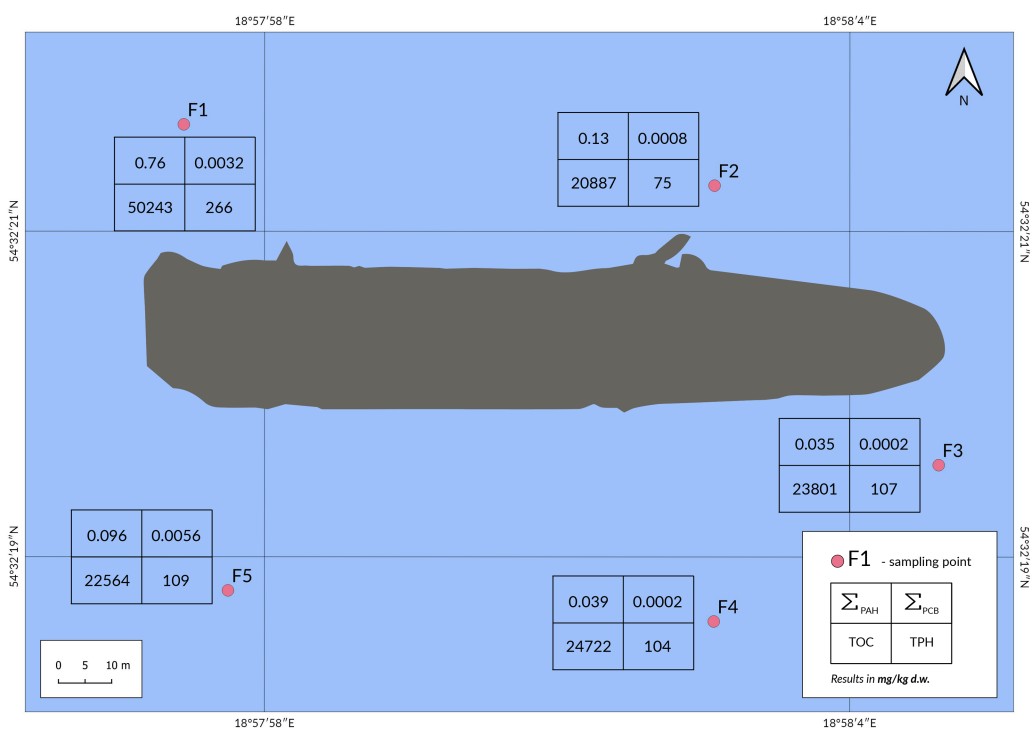

**Figure 5 Location of the sampling points in the Franken shipwreck area, together with the results obtained for the parameters tested.** Image credit: Karolina Duljas.

## RESULTS

### S/T Burgmeister Petersen

Analysis of four bottom sediment core samples taken in the vicinity of the S/T Burgmeister Petersen shipwreck (Fig. 3) showed that the TPH content at points SW and NE was below the LOQ of the method used (<5.0 mg/kg d.w.), while at point NW it was slightly above LOQ (7.0 mg/kg d.w.). The TPH content at point SE was 51 mg/kg d.w. The distribution of the concentrations of the sum of the seven PAHs in the samples was low, as was the TPH content. In the NE sample, the concentrations of all PAHs determined were below the LOQ (<0.001 mg/kg d.w.), and in the SW sample only the concentrations of benzo[a]anthracene (0.002 mg/kg d.w.) and indeno[1,2,3-cd]pyrene (0.001 mg/kg d.w.) were above the LOQ. The NW sample showed slightly higher concentrations of sum of seven PAHs, which was due to the presence of benzo[a]anthracene (0.010 mg/kg d.w.), benzo[b]fluoranthene (0.011 mg/kg d.w.) and benzo[k]fluoranthene (0.008 mg/kg). The highest concentration of the sum of the seven PAHs was recorded in sample SE (1.2 mg/kg d.w.), where the concentrations of the PAHs determined ranged from 0.018 mg/kg d.w. for dibenzo[a,h]anthracene to 0.47 mg/kg d.w. for benzo[a]anthracene.

### m/s Sleipner

Analysis of five bottom sediment core samples collected in the vicinity of the m/s Sleipner shipwreck (Fig. 4) showed that none of these samples were contaminated with TPH

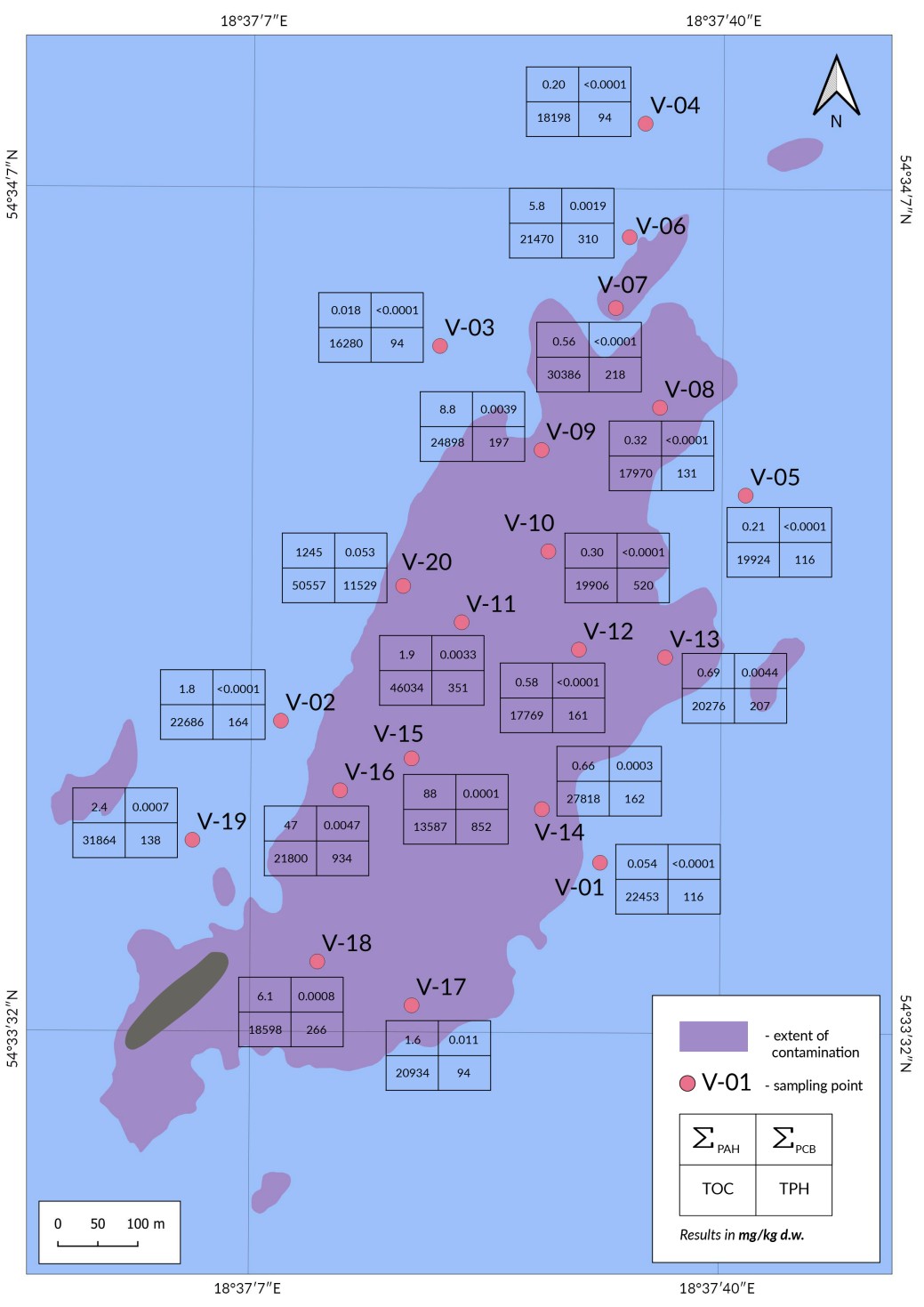

**Figure 6** Location of the sampling points in the Stuttgart shipwreck area, together with the results obtained for the parameters tested. Image credit: Karolina Duljas.

(<5.0 mg/kg d.w.) and PCBs (<0.0001 mg/kg d.w.). Benzo[a]anthracene (at all points), benzo[b]fluoranthene (at S1, S2, S4 and S5) and benzo[k]fluoranthene (at S2) were detected in the analysed samples, but these compounds were present at very low concentrations. The TOC content in all samples was below LOQ (<1,000 mg/kg d.w.).

### t/s Franken

Analysis of the five bottom sediment core samples showed that the total concentrations of the 7 PAHs were highest at point F1 (0.76 mg/kg d.w.) (Fig. 5). Moreover, the benzo[a]pyrene content was also the highest at this point (0.064 mg/kg d.w.) and strongly deviated from the results at the other points (0.002–0.006 mg/kg d.w.). It is worth noting that also in the case of TPH and TOC content, the results for the individual analyses are highest at point F1. The exception is the sum of PCBs, where their highest content was determined in point F5 (0.0056 mg/kg d.w.). The TOC content at point F1 was twice as high as the concentrations recorded at the other measurement points, where this parameter was at a similar level. The results obtained clearly indicate that point F1 diverges from the other study points in terms of concentrations of the sum of PAHs, TPH and TOC.

### S/s Stuttgart

Analysis of the twenty cored bottom sediments in the Stuttgart shipwreck area showed that the highest concentrations of the analysed analytes were found at point V-20 (Fig. 6). Points V-15 and V-16 showed elevated concentrations of the sum of PAHs and TPH compared to the other points analysed (in V-15 and V-16 ΣPAHs were 88 and 47 mg/kg d.w. respectively, TPH petroleum hydrocarbons 852 and 934 mg/kg d.w.). For the sum of PCBs, the second highest value was determined in V-17 (0.011 mg/kg d.w.), although this was still five times lower than in V-20 (0.053 mg/kg d.w.). High levels of TOC were recorded in bottom sediment core samples from all sampling points (13,587–50,557 mg/kg d.w.), even in those furthest from the wreck (points V-07, V-06 and V-04, where TOC concentrations were 30,386, 21,470 and 18,198 mg/kg d.w., respectively).

## DISCUSSION

The presented results of the studies conducted in the past years clearly indicate that the S/s Stuttgart is the biggest threat to the marine ecosystem among the investigated shipwrecks. Most of the samples taken from her surroundings contained high concentrations of the sum of PAHs, the sum of PCBs, TPH and TOC. This is most likely due to fuel spills caused by dismantle the shipwreck by pyrotechnic destruction (Dembska et al., 2016). In samples taken around the S/T Burgmeister Petersen, m/s Sleipner and t/s Franken shipwrecks, the tested parameters were at low levels. For these three shipwrecks, in none of the samples measured concentrations of PAHs, PCBs, TPHs and TOC exceeded values typical for sediments from these areas of the Baltic Sea and did not exceed the maximum permissible concentrations of PAHs and PCBs set by Polish legislation (Journal of Laws, 2015).

The results obtained from the investigations indicate that the m/s Sleipner shipwreck did not have a negative impact on the marine environment during the sampling period for the presented study. Despite the relatively low content of the analysed compounds in all tested

samples, it can be noted that at point SE in case of the S/T Burgmeister Petersen shipwreck and at point F1 in case of the t/s Franken shipwreck, the values obtained deviate from those determined at the other points. The area surrounding the S/T Burgmeister Petersen shipwreck and the m/s Sleipner shipwreck is dominated by erosion and transport activities of the sea, which contribute to the transfer of potential contaminants deep into the sea along with the sediments. In the case of the m/s Sleipner, this is evidenced by the short distance of the shipwreck from land and the very low TOC content (<1,000 mg/kg d.w.). In the case of the S/T Burgmeister Petersen shipwreck, no TOC analysis was performed, but the macroscopic description of the core samples indicates a total contribution of the fine sand fraction in each layer (*Dembska et al., 2011*). Few literature data are available on the contamination of sediments near non-urbanised areas of the Baltic Sea with petroleum compounds. The results of studies on sand samples from the southern Baltic coast conducted at the Lubiatowo Research Station show a TPH content below 1.0 mg/kg (*Otremba et al., 1996*). Studies performed by *Kaniewskim et al. (1996)* indicate that TPH concentrations in sediments located on the Polish territorial waters mostly do not exceed 1 mg/kg d.w. TPH content in sediments sampled on the Polish areas of the Baltic Sea rarely exceeds 25 mg/kg d.w., but depending on the location, these concentrations can be up to 600 mg/kg d.w. (*Sapota et al., 2012*). Total hydrocarbon content (THC) around archipelagos on the east coast of Sweden, on the other hand, range from 2 to 20 mg/kg d.w. (*Rudling, 1976*). These results are in line with those determined in most of the samples taken near the S/T Burgmeister Petersen and the m/s Sleipner shipwrecks. The exceptions are the elevated TPH and total PAH contents at the SE point near the S/T Burgmeister Petersen shipwreck, which were (51 mg/kg d.w.) and (1.2 mg/kg d.w.), respectively. This may indicate potential pollution of this site by fuel spillage from the shipwreck remains. In order to assess the source of the contamination at point SE, additional investigations of sediment samples would be necessary, primarily from the area between point SE and the shipwreck and from a location further away from the shipwreck than point SE. Elevated concentrations of the analysed compounds at point F1, on the other hand, may be due to the presence of strong bottom currents in the area of the t/s Franken (*Hac et al., 2016a*). The higher TOC values in the vicinity of the t/s Franken shipwreck than in the vicinity of the m/s Sleipner shipwreck may be due to differences in terrain. TOC content is low in coastal areas with coarse sediments and gradually increases with depth. These differences are due to wave motion and sea currents (*Jansen et al., 2003*; *Lubecki & Kowalewska, 2010*). The TOC content determined in samples from the south-western Baltic Sea ranged from 0.05–7.0% (*Christoffersen et al., 2007*; *Wang et al., 2020*). The total organic matter (TOM) content around the Gulf of Gdańsk varies from 4.6% in samples taken at a depth of 16 m to as much as 21.2% in samples taken at a depth of 89 m (*Pazdro, 2004*). The t/s Franken shipwreck is far from shore and therefore the sediment samples are dominated by a fine-grained fraction, with a better ability to accumulate organic matter. The results obtained may indicate the accumulation of not only organic matter of natural origin on the bottom, but also the gradual accumulation of compounds from the shipwreck. In 2018, investigations were carried out suggesting that partial contamination of the sediments near the t/s Franken shipwreck may have occurred. Investigations at the vicinity of the shipwreck

**Table 1** Range of concentrations of the sum of PAHs determined in the bottom sediments of the Baltic Sea–literature data and in bottom sediment core samples taken in the vicinity of the shipwrecks investigated - this study.

| Area | Number of PAHs analysed | ∑PAH mg/kg d.w. | Year of study | References |
|---|---|---|---|---|
| Burgmeister Petersen wreck area | 7 | 0.001–1.2 | 2011 | this study |
| Sleipner wreck area | 7 | 0.001–0.003 | 2016 | this study |
| Franken wreck area | 7 | 0.035–0.76 | 2016 | this study |
| Stuttgart wreck contamination area | 7 | 0.018–1245 | 2016 | this study |
| Belts Sea and Arkona Sea | 15 | 0.011–1.9 | – | *Witt & Trost (1999)* |
| Gulf of Gdańsk | 14 | 0.235–2.205 | 2002 | *Pazdro (2004)* |
| Coastal area of the Gulf of Gdańsk | 16 | 0.200–52.0 | 2002–2003 | *Falandysz et al. (2006)* |
| Gulf of Gdańsk | 12 | 0.009–5.1 | 2003–2007 | *Lubecki & Kowalewska (2010)* |
| Port basin of the Port of Gdynia | 7 | 0.677–9.0 | 2012 | *Pazikowska-Sapota et al. (2016)* |
| | 16 | 2.904–14.66 | | |
| Baltic Sea | 18 | 0.0009–5.361 | 2017 | *Wang et al. (2020)* |

revealed the presence of empty tanks from which fuel may have previously leaked as a result of an unsealing. In addition, total PAH content of 899–1,780 mg/kg d.w. was detected in the samples collected, which is significantly higher than the total PAH content in the samples tested in 2016. Furthermore, high concentrations of other compounds, including TPH and TOC, were also found. It should be noted that the publication does not provide precise data as to the location of the sampling points, only that they were in the vicinity of the t/s Franken shipwreck (*Hac, 2018*). It is therefore impossible to indicate what area around the ship may have been polluted.

While the content of individual analysed compounds in samples from the vicinity of the three shipwrecks remained at low levels, often not exceeding the LOQ, the results for samples from the S/s Stuttgart shipwreck area are completely different. Literature data show that the TOC content determined in samples taken in the vicinity of S/s Stuttgart does not differ from literature values for the Baltic Sea (*Christoffersen et al., 2007*; *Wang et al., 2020*), but due to the location of the shipwreck in the coastal zone a TOC content of less than 1% would be expected (*Szczepańska & Uścinowicz, 1994*). A study from 2004 showed that sediment samples from this part of the Gulf of Gdańsk consisted mostly of the sand fraction, with a TOM not exceeding 0.86% (*Pazdro, 2004*). The higher TOC content in samples V-1–V-20 taken from the vicinity of the S/s Stuttgart shipwreck may be due, among other reasons, to the silty and clayey nature of the collected sediments, which are not typical for this part of the Gulf of Gdańsk. However, the results indicate that the pollution of the environment by fuel has definitely increased the content of organic matter in this area. The level of contamination of seabed sediments with PAH group compounds is significantly higher in samples collected in the vicinity of the S/s Stuttgart shipwreck than in other studied samples from the Gulf of Gdańsk area (Table 1). According to literature data, the main area of PAH accumulation in the Gulf of Gdańsk is the Gdańsk Deep, where the sum of 12 PAHs in bottom sediments, for the surface layer (0–10 cm) was 3,600 ng/g d.w. (3.6 mg/kg d.w.) (*Lubecki & Kowalewska, 2010*). This is confirmed by studies conducted in the Gulf of Gdańsk in 2004, where the highest concentrations of the sum of 14 PAHs

were also recorded at two points located in the Gdańsk Deep: 1933.6 and 2205.0 ng/g d.w. (1.9 and 2.2 mg/kg d.w.) (*Pazdro, 2004*). The values determined in samples from the vicinity of the S/s Stuttgart shipwreck can also be compared with the results of studies of bottom sediments taken from the port area of the Port of Gdynia, which, as an area heavily influenced by human activity, show significant contamination with compounds from the PAH group. The concentrations determined in this area ranged from 0.677–9.0 mg/kg d.w. (sum of seven PAHs) and 2.904–14.66 mg/kg (sum of 16 PAHs) (*Pazikowska-Sapota et al., 2016*). Table 1 also presents the obtained ranges of concentrations of PAHs around the discussed shipwrecks. When analysing the above data, it may be noticed that the results achieved in many sampling points from the S/s Stuttgart shipwreck area, significantly exceed even those values which are characteristic for the most polluted areas of the Gulf of Gdańsk. In addition, it is worrying that the results of analyses of the sum of PAHs in six samples taken in the area of contamination around the S/s Stuttgart showed that the maximum permissible concentrations specified in the Regulation on recovery of waste outside installations and facilities (*Journal of Laws, 2015*) are exceeded (Table 2). Based on the results obtained and the analysis of the literature data, it can be concluded that the source of the bottom sediment contamination with PAHs in this area is the S/s Stuttgart shipwreck lying on the seabed. The results of the study and the conclusions drawn are consistent with the data presented in 2010 on the study of the environmental impact of the S/s Stuttgart shipwreck, which obtained results of a sum of 16 PAHs ranging from 0.686–1,291 mg/kg d.w. in bottom sediment core samples and 11.54–206.7 mg/kg d.w. in surface sediment samples (*Rogowska, Wolska & Namieśnik, 2010*). The concentrations of the indicators determined in samples taken in the vicinity of the S/s Stuttgart not only stand out from the standard values for this area, but also differ significantly from the values determined around other shipwrecks lying in the sea waters. *Van Landuyt et al. (2022)* analysed samples from the vicinity of the World War II wreck V-1302 John Mahn. Despite the determination of 23 PAH compounds in these samples, the determined values of total PAHs were below 250 µg/kg (0.25 mg/kg). Studies of samples taken in the vicinity of HMS Royal Oak, which was sunk during World War II off the coast of Scotland, showed PAHs and their alkylated derivatives to be no higher than 350 µg/kg d.w. (0.35 mg/kg d.w.) (*Thomas et al., 2021*). In 2017, bottom sediment samples collected in the vicinity of the Nordvard ship that sank off the coast of Norway were analysed. The maximum determined concentration of 16 PAHs in these samples was less than 30 mg/kg, while the maximum TPH content did not exceed 8 g/kg (8,000 mg/kg) (*Ndungu et al., 2017*). In 2018, there was a collision between the Sanchi tanker and the Changfeng Crystal ship on the South China Sea. Analyses performed one month after the accident showed the content of the sum of 16 PAHs in the bottom sediment at a maximum level of 227 ng/g (0.227 mg/kg) (*Yin et al., 2021*).

While the results for PAH content at individual points are understandable, the concentrations of the sum of PCBs, especially at points V-17 and V-20 in the vicinity of the S/s Stuttgart Shipwreck, may seem puzzling. Although the PCB concentrations at these points stand out from the other results, they do not deviate from the available literature values for Baltic Sea bottom sediments (Table 3). In addition, the values determined in the

**Table 2** Results of individual PAHs in sediment core samples taken in the vicinity of S/s Stuttgart that exceed the maximum permissible concentrations set out in *Jounal of Laws (2015)* (mg/kg d.w).

| PAHs analysed | V-06 | V-09 | V-15 | V-16 | V-18 | V-20 |
|---|---|---|---|---|---|---|
| Benz[a]anthracene | **≥1,5** | **≥1,5** | **≥1,5** | **≥1,5** | **≥1,5** | **≥1,5** |
| Benzo[b]fluoranthene | <1,5 | **≥1,5** | **≥1,5** | **≥1,5** | **≥1,5** | **≥1,5** |
| Benzo[k]fluoranthene | <1,5 | <1,5 | **≥1,5** | **≥1,5** | <1,5 | **≥1,5** |
| Benzo[a]pyrene | <1,0 | **≥1,0** | **≥1,0** | **≥1,0** | **≥1,0** | **≥1,0** |
| Indeno[1,2,3-cd]pyrene | <1,0 | **≥1,0** | **≥1,0** | **≥1,0** | <1,0 | **≥1,0** |
| Dibenz[a,h]anthracene | <1,0 | **≥1,0** | **≥1,0** | **≥1,0** | <1,0 | **≥1,0** |
| Benzo[g,h,i]perylene | <1,0 | <1,0 | **≥1,0** | **≥1,0** | <1,0 | **≥1,0** |

**Notes.**

Values in bold indicate those exceeding the maximum permissible concentrations set out in *Jounal of Laws (2015)*.

**Table 3** Summary of PCB concentrations determined in the bottom sediments of the Baltic Sea—literature data and in bottom sediment core samples taken in the vicinity of the shipwrecks investigated—this study. Abbreviation: n.d., no data.

| Area | Number of PCBs analysed | ΣPCBs (7 congeners) range (mg/kg d.w.) | Year of study | References |
|---|---|---|---|---|
| Stuttgart wreck contamination area | 7 | <0.0001–0.0530 | 2016 | this study |
| Burgmeister Petersen wreck area | 7 | n.d. | 2011 | this study |
| Sleipner wreck area | 7 | <0.0001 | 2016 | this study |
| Franken wreck area | 7 | 0.0002–0.0056 | 2016 | this study |
| Gulf of Gdańsk | 7 | 0.0029–0.078 | 2002 | *Pazdro (2004)* |
| Transitional zone between the North Sea and the Baltic Sea | 12 | 0.100–0.989 | – | *Christiansen et al. (2009)* |
| Gulf of Bothnia | 68 | 0.009–0.0093 | 1991 | *Strandberg et al. (2000)* |
| The western Baltic | 23 | 0.0002–0.0087 | 1993–1994 | *Dannenberger & Lerz (1996)* |
| The Gdańsk Deep | 7 | 0.020–0.148 | 1999 | *Konat & Kowalewska (2001)* |
| Gulf of Gdańsk | 7 | 0.022–0.033 | 1999 | *Konat & Kowalewska (2001)* |
| Gulf of GdańskStuttgart wreck area | 7 | 0.0008–0.1088 | 2009 | *Rogowska, Wolska & Namieśnik (2010)* |
| Gulf of GdańskStuttgart wreck area | 7 | 0.004–0.354 | – | *Rogowska et al. (2015)* |

samples taken in the vicinity of the S/s Stuttgart do not exceed the maximum permissible PCB concentrations set out in OJ of 2015, item 796 (<0.3 mg/kg d.w.). Surveys carried out in earlier years in the vicinity of the S/s Stuttgart shipwreck showed two sampling points with outliers in terms of the sum of PCBs: 354 and 151 ng/g d.w. (0.354 and 0.151 mg/kg d.w.) (*Rogowska et al., 2015*). According to *Rogowska et al. (2015)* the reason for this may be illegal discharges of PCB-containing material. It is worth to mention that, according to *Falandysz (1999)*, PCBs were used not only in transformer oils and small capacitors, such as washing machines and refrigerators, but also as an additive to petroleum products or as a softener for the insulation of electrical cables and wires. In addition, from at least the 1830s, XBayer AG in Germany produced Clophen A, which was available commercially as a technical preparation containing a mixture of different PCB congeners (*Falandysz, 1999*). Compositional studies conducted in the 1980s on several commercially available Clophen A preparations confirmed the presence of most of the determined, by the Maritime Institute,

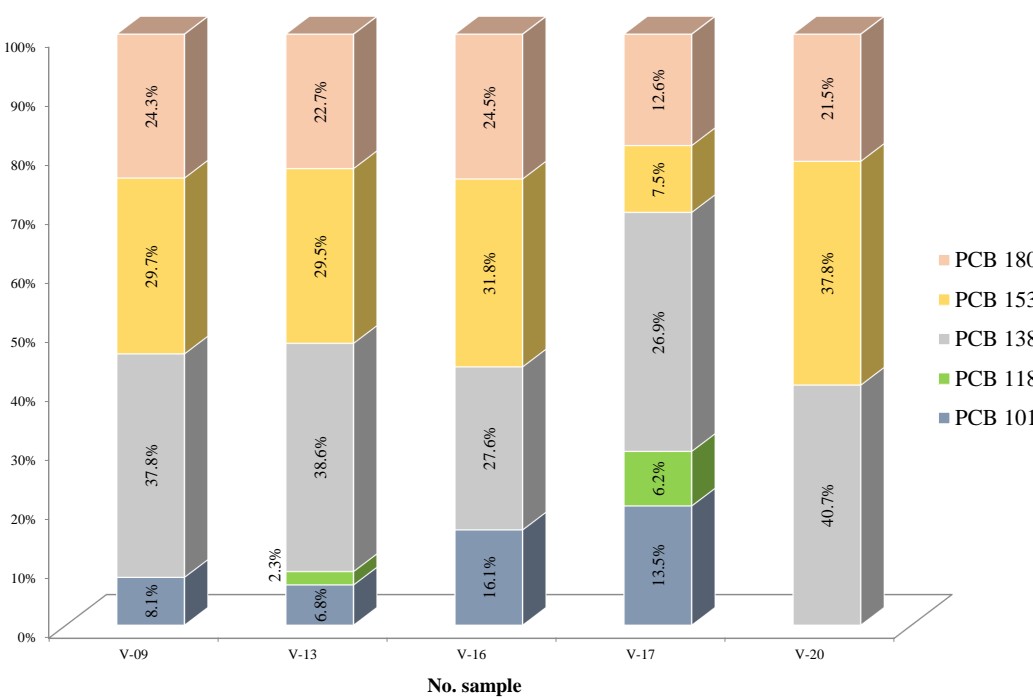

**Figure 7** Percentage content of individual PCB congeners determined in selected core sediment samples collected from the Stuttgart shipwreck area.

PCBs in core samples (*Ballschmiter & Zell, 1980*; *Schulz, Patrick & Duinker, 1989*). It is therefore possible that Clophen A was used as a technical preparation in some equipment on board the S/s Stuttgart. All this may lead to the conclusion that the bottom sediment core samples were contaminated with PCB congeners precisely as a result of the dismantle of the S/s Stuttgart shipwreck by pyrotechnic destruction. It should be stressed that even in sample V-20 and V-17, where the results for PCBs were the highest (0.053 and 0.011 mg/kg d.w.) the values obtained are lower than the highest obtained by *Rogowska et al. (2015)*. This is most likely due to the sampling location: the points tested in 2015 are located at a distance of tens of metres from the shipwreck, while points V-17 and V-20 are located at a distance of several hundred metres. The occurrence of the highest PCB concentrations in the vicinity of the shipwreck, which decrease with increasing distance, confirm the assumption that PCB congener contamination originates from S/s Stuttgart.

The percentage composition of PCB congeners in the five selected sampling points in the vicinity of the S/s Stuttgart shipwreck, where the highest PCB sum was determined, is shown in Fig. 7. In all the core sediment samples shown, a similar percentage distribution of PCB congeners can be observed. The absence of PCB congeners no. 28 and 52, as well as the low percentages of congeners no. 101 and 118 in relation to the sum of PCBs, may be due to better solubility in water. Some studies additionally suggest that PCB degradation

is possible, especially for congeners having five or fewer chlorine atoms (*Carey & Harvey, 1978*; *Brownawell & Farrington, 1986*; *Falandysz, 1999*).

Pollution of the Baltic Sea with petroleum substances is mainly due to anthropogenic sources. Although pollution resulting from shipwrecks has the most tragic consequences, the cause of cyclic oil pollution of the sea is to be found in river run-off. It is estimated that 21–66,000 tonnes of oil-derived substances enter the Baltic Sea annually (*Szymelfenig & Urbański, 1998*; *Fabisiak, 2008*). Although the content of oil substances is a significant threat to the southern parts of the Baltic Sea, due to the large catchment areas, there are still limited publications on their content in sediment. A study by *Kaniewskim et al. (1996)* shows that hydrocarbon concentrations in sediments in the Gulf of Gdańsk, depending on the location, can be as high as 100 mg/kg d.w. Similar results were obtained by *Law & Andrulewicz (1983)* when studying bottom sediments of the southern Baltic Sea. THC values ranged from 4.0 mg/kg d.w. in the Gulf of Gdańsk to 140.0 mg/kg d.w. in the Gotland Basin. Data from other urbanised areas of the Baltic Sea show even higher results. For example, sediments taken off the coast of Sweden contained THC in the range of 40–400 mg/kg d.w. (*Rudling, 1976*). Generally, in sediments from the Polish part of the Southern Baltic, TPH content is found at levels up to 25 mg/kg d.w. In bottom sediments of Polish ports, TPH content ranges from 300 to 625 mg/kg d.w. (*Sapota et al., 2012*). These results are in line with those obtained from samples in the t/s Franken shipwreck area and suggest her minor influence on sediment contamination by petroleum substances. Analytical results of some samples taken from the vicinity of the S/s Stuttgart shipwreck (points V-15–852 mg/kg d.w.; V-16–934 mg/kg d.w. and V-20–11,529 mg/kg d.w.) deviate from literature values and may indicate point-in-time increased contamination of the seabed. Higher TPH concentrations were determined, *e.g.*, in sediment samples collected by *Suzdalev & Gulbinskas (2014)* in the Klaipeda Strait (Lithuania). In most of these samples, the TPH content was below 100 mg/kg d.w., but at several areas, mainly bordering port zones, the TPH content was in the range of 200–1,500 mg/kg d.w., and at one point it exceeded 1,500 mg/kg d.w. Previous studies in the Klaipeda Port area indicated a TPH content of 2,029 mg/kg in clay sediments (*Stakeniene, 1999*; *Suzdalev & Gulbinskas, 2014*). Such high values are due to the fact that the studies were carried out on a strait which, although it has constant water circulation, is an area of intense accumulation (*Trimonis, Vaikutiene & Gulbinskas, 2010*; *Stakeniene, Galkus & Joksas, 2011*; *Suzdalev & Gulbinskas, 2014*).

There are currently no Polish legal standards specifying a limit value for TPH content indicating bottom sediment contamination, but such legal regulations exist in Baltic countries such as Estonia, Finland, Germany, Latvia, Lithuania and Russia. Most of these countries set two limit values (first and second limit value). If the content of hazardous substances in the sediment does not exceed the first limit value, the sediment is considered clean and can be stored at sea. In contrast, a content of hazardous substances in the dredged sediment above the second limit value indicates that the sediment is contaminated and cannot be stored at sea. If the content of contaminants is between the first and second limit value, such sediment is defined as potentially contaminated and a decision on its storage can be taken after additional testing. Examples of TPH limit values applicable in *e.g.*, Latvia are 100 mg/kg d.w. (first limit value) and 400 mg/kg d.w. (second limit value),

and in Finland 50 mg/kg d.w. (first limit value) and 1,500 mg/kg d.w. (second limit value) (*Sapota et al., 2012*). If the TPH values determined in the samples taken in the vicinity of the S/s Stuttgart shipwreck were referred to the Latvian regulations, samples from only three points would be considered clean and samples from four points would be considered contaminated. If Finland's regulations were referred to, one sample would be found to be contaminated, while the remaining 19 samples are potentially contaminated. The results obtained clearly indicate that the bottom sediments around the S/s Stuttgart shipwreck have the potential to contaminate the Baltic Sea.

## CONCLUSIONS

The studies conducted indicate that the S/s Stuttgart shipwreck is the biggest threat to the marine ecosystem of the Baltic Sea. Most of the samples taken from her vicinity showed significantly elevated levels of the sum of PAHs, TPH and TOC. Further work would therefore need to be undertaken to minimise the negative impact of pollution from this shipwreck on the marine environment. The results of the core sediments taken in the vicinity of the t/s Franken and S/T Burgmeister Petersen shipwrecks are puzzling. Elevated concentrations of analysed compounds may indicate a potential fuel spill from these ships. Surveys carried out two years later near the t/s Franken shipwreck also indicate a probable fuel leak (*Hac, 2018*). To confirm this, the surveys around both shipwrecks should be extended to include further sampling points for analysis. Based on the results of the analyses of the samples taken around the m/s Sleipner shipwreck, it can be concluded that there was no fuel leak. However, these surveys were carried out in 2016 and, therefore, monitoring of the area around this ship would need to be carried out again to confirm this.

For many years it was thought that shipwrecks resting on the Baltic Sea seabed did not pose a major threat to the marine environment. However, it should be highlighted that each shipwreck is a separate component of this complex system and is a potential hazard. Fuels in the shipwreck's tanks may escape into the marine environment due to progressive corrosion. It is therefore good practice to protect the Baltic Sea ecosystem to prevent environmental disasters in advance by monitoring the areas around all shipwrecks and classifying them to determine potential risk.

## ACKNOWLEDGEMENTS

The authors are grateful for the support of Jacek Koszałka, Karolina Duljas and Justyna Edut.

### Funding

This research was funded by "Wrecks as a source of marine environment pollution", grant number IM/2023/PZ/02. The funders had no role in study design, data collection and analysis, decision to publish, or preparation of the manuscript.

### Grant Disclosures

The following grant information was disclosed by the authors:
Wrecks as a source of marine environment pollution: IM/2023/PZ/02.

### Competing Interests

The authors declare there are no competing interests.

### Author Contributions

- Agata Szpiech conceived and designed the experiments, performed the experiments, analyzed the data, performed the computation work, prepared figures and/or tables, authored or reviewed drafts of the article, and approved the final draft.
- Aleksandra Bojke conceived and designed the experiments, analyzed the data, performed the computation work, prepared figures and/or tables, authored or reviewed drafts of the article, and approved the final draft.
- Małgorzata Littwin performed the experiments, analyzed the data, authored or reviewed drafts of the article, and approved the final draft.
- Aleksandra Dzendrowska performed the experiments, analyzed the data, performed the computation work, prepared figures and/or tables, authored or reviewed drafts of the article, and approved the final draft.
- Ewelina Duljas performed the experiments, analyzed the data, performed the computation work, prepared figures and/or tables, authored or reviewed drafts of the article, and approved the final draft.
- Agnieszka Flasińska performed the computation work, authored or reviewed drafts of the article, and approved the final draft.
- Katarzyna Szczepańska performed the experiments, prepared figures and/or tables, and approved the final draft.
- Tomasz Dziarkowski performed the experiments, prepared figures and/or tables, and approved the final draft.
- Grażyna Dembska conceived and designed the experiments, authored or reviewed drafts of the article, and approved the final draft.
- Grażyna Pazikowska-Sapota performed the computation work, authored or reviewed drafts of the article, and approved the final draft.
- Katarzyna Galer-Tatarowicz conceived and designed the experiments, authored or reviewed drafts of the article, and approved the final draft.
- Benedykt Hac performed the experiments, authored or reviewed drafts of the article, and approved the final draft.

### Data Availability

The raw data are available in the Supplemental File.

### Supplemental Information

Supplemental information for this article can be found online at http://dx.doi.org/10.7717/peerj-achem.31#supplemental-information.

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
