# Peer review of "Baltic Sea shipwrecks as a source of hazardous pollution"

_PeerJ Analytical Chemistry, doi:10.7717/peerj-achem.31_

## Round 0.1 · original submission · Major Revisions

Dear Authors,
Reviewers have evaluated your manuscript. it is recommended that you revise the manuscript to polish the draft to a standard that is acceptable in a journal with international audience. In addition to this, please consider the following additional EDITOR comments:

1. The abstract needs to be carefully revised following the journal guidelines found in the template at https://peerj.com/about/author-instructions/chemistry. Specifically, you need to properly introduce the study background (problem), present the methods used and the key findings (results).

2. Some of the literature cited are too old. For example, Andrulewicz et al. (1994, 1998), (HELCOM, 1974), and (Jeffery, 1990). Although some of these may be important in this study, such literature rarely present the current status of knowledge and should only be cited in exceptional cases. There is very recent relevant literature in this area.

https://doi.org/10.3390/w14223772
https://doi.org/10.1016/j.marpolbul.2021.112747
https://doi.org/10.1016/j.marenvres.2020.105036
https://doi.org/10.1016/j.marpolbul.2022.114426

3. The introduction needs to be rewritten altogether so that it does not merely present the history of the ship wreckages that occurred in the study area. This section should introduce the potential of ship wrecks as sources of pollution (see for example, https://doi.org/10.3390/jmse11020276), which should better position your study and justify why you need to report on the study area at this point in time.

4. Thoroughly check the manuscript for typos and grammar. For example, in L157, ''or'' should be ‘‘for’’.

5. Methodology (L156-178) needs to be elaborated. The analytical conditions used should be clearly indicated, including any quality control and quality assurance activities undertaken. See https://doi.org/10.1007%2Fs40201-019-00356-z for a similar reporting.

6. Your study does not have any dedicated section on how the data were analyzed or visualized. I believe that your study could benefit from health risk assessment, which should give a better view as to whether the current pollution levels pose any significant health threats to the local pollution.

7. In Table 3, it is not possible that Pazdro (2004) did their study in 2022. The ‘‘n.d.’’ in this Table should be replaced with the LOD or LOQ. For easy comparison, put the results of your study first, followed by previous studies.

Reviewer 1 ·

Basic reporting

no comment

Experimental design

no comment

Validity of the findings

no comment

Additional comments

See comments in attached PDF

Annotated reviews are not available for download in order to protect the identity of reviewers who chose to remain anonymous.
Cite this review as
Anonymous Reviewer (2024) Peer Review #1 of "Baltic Sea shipwrecks as a source of hazardous pollution (v0.1)". PeerJ Analytical Chemistry

Reviewer 2 ·

Basic reporting

no comment

Experimental design

no comment

Validity of the findings

no comment

Additional comments

Line 32: Ocean should be specified: Northe Sea/Atlantic
Line 34: deterioration instead of degradation should be better used
Line 47: insert new paragraph
Line 81: insert new paragraph
Legend to Fig. 7 does not fit to Fig. 7
Line 157: "...were analysed for 7 polycyclic aromatic ..."
Line 445: please add "in the Baltic Sea"

Cite this review as
Anonymous Reviewer (2024) Peer Review #2 of "Baltic Sea shipwrecks as a source of hazardous pollution (v0.1)". PeerJ Analytical Chemistry

---

## Round 0.2 · accepted · Accept

The authors have made great improvements on the previous version of the manuscript. Your manuscript is now in a version that I believe will be of interest to the audience of PeerJ Analytical Chemistry

Reviewer 2 ·

Basic reporting

No further comment

Experimental design

no comment

Validity of the findings

no comment

Additional comments

I have no further comments

Cite this review as
Anonymous Reviewer (2024) Peer Review #2 of "Baltic Sea shipwrecks as a source of hazardous pollution (v0.2)". PeerJ Analytical Chemistry